# MQSP: Micro-Query Sequence Parallelism for Linearly Scaling Long Sequence Transformer

## Abstract

Long sequence modeling of Transformer gains prevalence in fields involving long texts and high-resolution images and videos but suffers from quadratic memory complexity. Existing work investigates low-complexity variants or parallel methods to handle it. The former attempts to approximate full attention and is limited by a single device's capacity. The latter struggles to manage quadratic memory of attention maps, leading to insufficient sequence scalability. In this work, we propose a novel parallel method named **M**icro-**Q**uery **S**equence **P**arallelism. MQSP slices sequences across devices and projects local queries, keys, and values in self-attention. For communication and memory efficiency, MQSP all-gathers the queries while keys and values remain locally to acquire the local attention map, on which a distributed softmax gets conducted to amortize memory by column. Meanwhile, the queries get further partitioned as Micro-Q to divide the computation and recycle the attention map by row, jointly decomposing the quadratic memory to achieve linear scalability. The evaluation result shows that MQSP scales up sequence length linearly and achieves $4.5\times$ sequence length of ColossalAI's sequence parallelism and $4.3\times$ of Megatron-LM3, enabling training BERT-large of 78848 sequence length on 32 A100 GPUs. MQSP can reduce up to $78.6\%$ of memory occupation and achieve up to $3.3\times$ throughput when training on 17408 sequence length. The convergence quality experiment proves that MQSP provides the means for long sequences with guaranteed convergence, bringing the potential for the Transformer to explore longer sequences.

## 1 Introduction

Transformer (Vaswani et al., 2017), an attention-based model initially proposed for natural language processing (NLP), shows its promising potential in computer vision (CV), multi-modality, and more (Carion et al., 2020; Dosovitskiy et al., 2021; Liu et al., 2022; Arnab et al., 2021; Neimark et al., 2021; Radford et al., 2021; Wang et al., 2022). The self-attention associations between arbitrary pairs of tokens enable the Transformer to learn global context-aware sequence representation for many modalities. Furthermore, there is an emerging trend toward long-range modeling, which scales up the sequence length of the Transformer. Long-range modeling is essential for the long texts in question answering, document classification, and other NLP tasks, as well as the high-resolution pictures and the series of video frames in image modality.

As the sequence gets extended, memory consumption increases rapidly due to the quadratic complexity of self-attention, inevitably exceeding the limit of a single device (e.g., GPU). This problem obstructs the exploration of the Transformer for modeling longer sequences. Recently researchers focused on sparse mechanisms in self-attention (Child et al., 2019; Beltagy et al., 2020; Zaheer et al., 2020) or low-complexity substitutes (Wang et al., 2020; Xiong et al., 2021; Choromanski et al., 2021; Qin et al., 2022) to approximate full attention, boosting sequence length. However, besides concern about performance influence, the memory upper bound of a single device still limits those methods from scaling up further.

Much existing work investigates parallel methods to distribute the Transformer model across the devices, such as tensor parallelism (Shoeybi et al., 2019), pipeline parallelism (Harlap et al., 2018; Huang et al., 2019; Narayanan et al., 2021; Yang et al., 2022), and zero redundancy optimizer (ZeRO) (Rajbhandari et al., 2020; Ren et al., 2021). However, these methods separate the model parameters

across different dimensions, not alleviating the enormous intermediate activations introduced by long sequence self-attention.Therefore, several more recent methods are devoted to partitioning Transformer along the sequence dimension. ColossalAI's sequence parallelism (Li et al., 2021) transfers keys and values in ring-style to compute the partial attention map. Megatron-LM3 (Korthikanti et al., 2022) modifies the conjugate operators to parallel layer-norms and dropouts in sequence dimension. Despite amortizing some intermediate activations, these approaches still consume local memory proportional to global sequence length, making it difficult to scale up to a longer sequence.

In this paper, we propose MQSP, a novel sequence parallel method, to diminish the quadratic memory overhead and efficiently scale up the long sequence Transformer. MQSP splits the input sequence to $n$ devices for parallel computation and projects local queries, keys, and values in self-attention. We design a distributed self-attention to handle the global context-awareness in self-attention. Specifically, MQSP synchronizes the queries across the devices through an all-gather operation while the keys and values remain local, acquiring local attention maps amortized along the column dimension. Since the rows for softmax are incomplete locally, we conduct a distributed softmax with hierarchical reduction across the devices, which introduces negligible communication. More importantly, we divide local queries into $m$ finer-grained queries, called Micro-Q, and process them step by step to get the corresponding attention outputs. Each micro step's memory space for the attention map would be shared along the row dimension, jointly decomposing the quadratic memory for linearly scaling up the sequence length.

The proposed MQSP shows significant sequence scaling ability compared with the existing leading approaches. Our evaluation indicates that for the Transformer BERT-large (Kenton & Toutanova, 2019), MQSP could scale up to 78848 sequence length with 32 A100 GPUs, $4.5\times$ of ColossalAI's sequence parallelism, and 38912 sequence length with 16 A100 GPUs, $4.3\times$ of Megatron-LM3. In memory usage and throughput comparison, MQSP saves 78.6% memory and achieves $3.3\times$ speedup, demonstrating the superiority of its distributed attention and communication method. Furthermore, the convergence quality experiments on wikitext, SQuAD, QQP and MRPC datasets prove that MQSP maintains the convergence quality for scaling up sequence length, bringing the prospect of exploring longer sequences to Transformers.

## 2 RELATED WORK

This section briefly introduces the self-attention complexity and the parallel methods for Transformer.

**Self-Attention Complexity.** In the Transformer proposed by Vaswani et al. (2017), self-attention is the vital module for the global dependencies modeling. Omit the batch and multi-head dimensions for brevity. For the input hidden states $x \in \mathbb{R}^{L \times d_m}$, where $L$ is the sequence length and $d_m$ is the model hidden size, the attention mechanism can be formulated as:

$$Q, K, V = \mathcal{L}_{qkv}(x), \quad C = \text{softmax}(S)V = \text{softmax}(\frac{QK^\mathsf{T}}{\sqrt{d_k}})V \tag{1}$$

Where $\mathcal{L}_{qkv}$ is the linear layer projecting the tokens to $Q, K \in \mathbb{R}^{L \times d_k}$, and $V \in \mathbb{R}^{L \times d_v}$ in the query, key, and value embedding spaces. The scaled dot product of $Q$ and $K$ produces the attention scores map $S \in \mathbb{R}^{L \times L}$, which incurs the quadratic complexity $O(L^2)$. Subsequently, the softmax operation along the rows converts $S$ to attention probabilities $P$, which reweights $V$ to the context output $C$.

Many recent approaches introduce sparse mechanisms, such as Sparse Trans. (Child et al., 2019), Longformer (Beltagy et al., 2020), and BigBird (Zaheer et al., 2020), or low-complexity substitutes, such as Linformer (Wang et al., 2020), Nyströmformer (Xiong et al., 2021), Performer (Choromanski et al., 2021), and Cosformer (Qin et al., 2022). These methods algorithmically approximate the full attention with sparsity or low-rank assumption, reducing complexity to $O(LlogL)$ or $O(L)$. However, unsatisfied assumptions could meet performance deficiency in a broad task spectrum, and accommodating the entire Transformer within one device still limits the further expansion of the sequence length. Thus we set our sights on scaling up standard Transformer through more devices.

**Parallel Methods for Transformer.** Parallelism approaches have been the innovative techniques for training the large Transformer. Pipeline parallelisms (Harlap et al., 2018; Huang et al., 2019; Narayanan et al., 2021; Yang et al., 2022) split the model layerwise without handling self-attention within layers. ZeRO (Rajbhandari et al., 2020) spreads the model's parameters and the optimizer states and conducts the same computation. Tensor parallelism in Megatron-LM (Shoeybi et al., 2019)

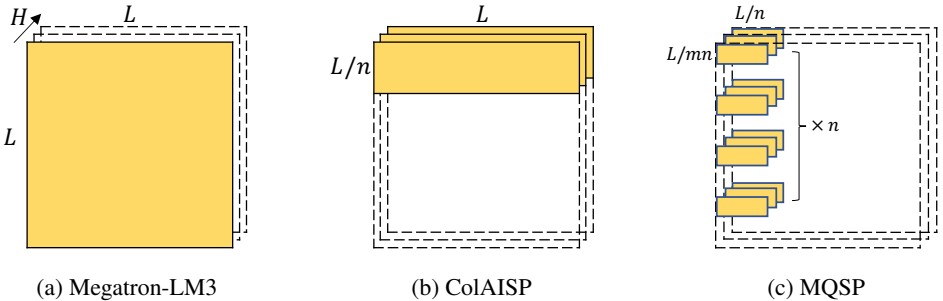

Figure 1: The attention maps for (a) Megatron-LM3, (b) ColossAI sequence parallelism, and (c) the proposed MQSP, with memory complexity of $O(\frac{H}{n}L^2)$, $O(H\frac{L^2}{n})$, and $O(H\frac{L^2}{mn})$.

decomposes the continuous linear layers and divides self-attention along multi-heads dimension (Fig. 1a). The methods above deal with model parameters but not the tremendous intermediate activations in self-attention that are quadratic to the sequence length, insufficient to scale up the sequence.

Therefore, slicing along the sequence dimension comes into mind for scaling long sequence Transformer. Intuitively, the input sequence $x$ could be split into $n$ chunks, $x \rightarrow \{x_0, x_1, .., x_{n-1}\}$, where $x_i \in \mathbb{R}^{\frac{L}{n} \times d_m}$, and fed into $n$ devices to compute in parallel. With similar insight, ColossalAI (Bian et al., 2021) proposed sequence parallelism (Li et al., 2021), or ColAISP for short. Considering the global associations of local query/key/value embeddings, which are projected in self-attention as $Q_i, K_i, V_i = \mathcal{L}_{qkv}(x_i)$, ColAISP designs Ring Self-Attention as:

$$C_i = \text{softmax}(S_{i,:})V = \text{softmax}(\frac{Q_i \overbrace{\left[K_0^\mathsf{T}, K_1^\mathsf{T}, ..., K_{n-1}^\mathsf{T}\right]}^{\textbf{RingQK}}}{\sqrt{d_k}}) \overbrace{\left[V_0^\mathsf{T}, V_1^\mathsf{T}, ..., V_{n-1}^\mathsf{T}\right]^\mathsf{T}}^{\textbf{RingAV}} \quad (2)$$

By ring-style transferring $K_i$ and $V_i$, ColAISP circularly computes $Q_i K_j^\mathsf{T}$ to collect the partial attention scores map $S_{i,:} \in \mathbb{R}^{\frac{L}{n} \times L}$(Fig. 1b). It indicates that ColAISP requires quadratic device resources $n$ to scale up $L$. Additionally, the efficiency of ring communication suffers from the weakest link, e.g., inter-node bandwidth. Megatron-LM3 (Korthikanti et al., 2022) modifies its conjugate operators to all-gather/reduce-scatter to introduce sequence parallelism in layer-norms and dropouts, yet leaves the self-attention in tensor parallel mode, resulting in the same quadratic attention maps.

## 3 METHOD

This section introduces the proposed Micro-Query Sequence Parallelism. We analyze the communication patterns in self-attention and propose Micro-Q for reused memory. Then the distributed softmax is described. Furthermore, we compare memory usage with other methods and analyze scalability.

### 3.1 MICRO-QUERY SEQUENCE PARALLELISM

Instead of dividing model parameters along different dimensions in previous parallel methods, we focus on the sequence dimension that affects intermediate activations. When the input sequences are partitioned and fed to devices in parallel, most modules, such as multi-layer perceptron (MLP), dropout, and layer normalization, are computed independently of the sequence dimension. However, as illustrated in Fig. 2, self-attention across devices does not natively support sequential parallelism. In MQSP, we dive into the self-attention mechanism to attain distributed attention with efficient communication and memory usage.

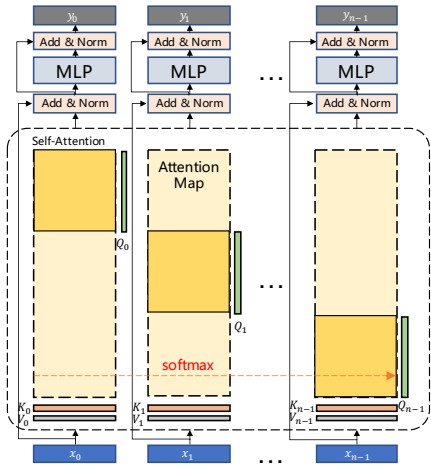

Figure 2: Illustration of sequence parallelism. Besides self-attention with global associations and softmax, other modules natively support sequence parallelism.

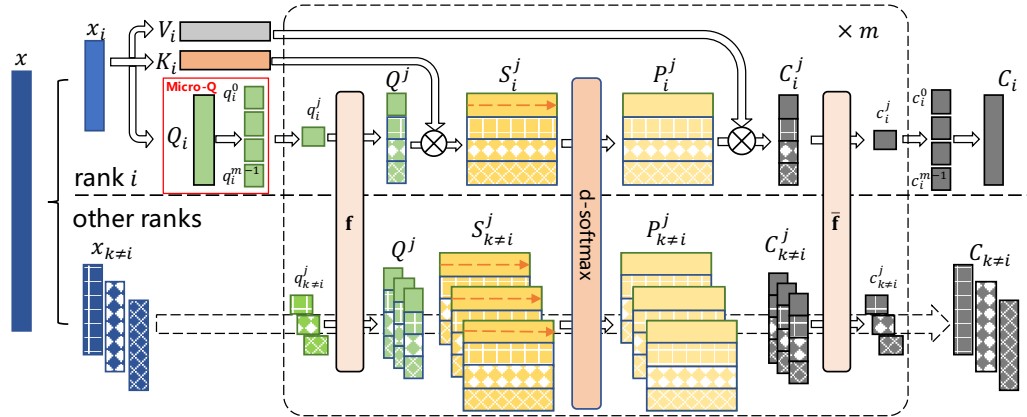

Figure 3: Overview of the MQSP self-attention. $\mathbf{f}$ and $\bar{\mathbf{f}}$ are the conjugate all-gather/reduce-scatter communication operations. The dotted box for $m$ times would reuse memory space. The red dotted arrow lines represent the aligned row for distributed softmax.

**Attention Analysis.** Recalling the general form of self-attention in Eq. 1, the self-attention for each query can be analyzed independently. Assuming the $k$-th query in $Q$ is $q_k \in \mathbb{R}^{1 \times d_k}$, the corresponding context $c_k$ is calculated as:

$$c_k = \text{softmax}(S_{row_k,:})V = \text{softmax}(\frac{q_k K^\intercal}{\sqrt{d_k}})V \qquad (3)$$

Where $S_{row_k,:} \in \mathbb{R}^{1 \times L}$ represents the $k$-th row of the attention scores map. ColAISP transfers $K_i$ as a ring and multiplies with $q_k$ to collect the complete $S_{row_k,:}$, which encounters efficiency deficiency in a bandwidth-imbalanced environment and includes the $L$ factor of memory overhead. In contrast, MQSP chooses to share the $q_k$ across the devices, remaining $K_i$ and $V_i$ locally. In this way, each rank only needs to compute $S_{row_k,i} \in \mathbb{R}^{1 \times \frac{L}{n}}$, its corresponding columns of attention scores:

$$c_k = \sum_{i=0}^{n-1} \text{d-softmax}(S_{row_k,i})V_i = \bar{\mathbf{f}}(\text{d-softmax}(\frac{\mathbf{f}(q_k)K_i^\intercal}{\sqrt{d_k}})V_i) \qquad (4)$$

Where d-softmax means the distributed softmax operation to acquire the corresponding columns of attention probabilities, further described in section 3.2. For the communications required here, we define $\mathbf{f}$ as $q_k$ synchronization across devices and $\bar{\mathbf{f}}$ as locally reweighted $V_i$ reduce-summation, e.g., broadcast/reduce or all-gather/reduce-scatter, as shown in Fig. 4. $\mathbf{f}$ and $\bar{\mathbf{f}}$ are conjugate, which means the forward pass of one equals the backward pass of the other. Considering the symmetric workload across devices and the efficiency of the duplex collective communication, we adopt all-gather/reduce-scatter as the conjugate operators to demonstrate our method (Fig. 3).

**Micro-Q.** The complete attention includes the whole sequence $q_{k \in [0,L)}$ in Eq. 4, leading to the equally significant attention scores $S_{:,i} \in \mathbb{R}^{L \times \frac{L}{n}}$. To this end, we propose Micro-Q, the finer-grained query, to reduce memory consumption orthogonally. For a number $m$ of Micro-Q, the local query $Q_i$ gets chunked as $\{q_i^0, ..., q_i^{m-1}\}$, as depicted in the red box of Fig. 3, where $q_i^j \in \mathbb{R}^{\frac{L}{mn} \times d_k}$ would be the $j$-th micro-step's input on the $i$-th device. The following $\mathbf{f}$ all-gathers $q_{i \in [0,n)}^j$ to form the shared

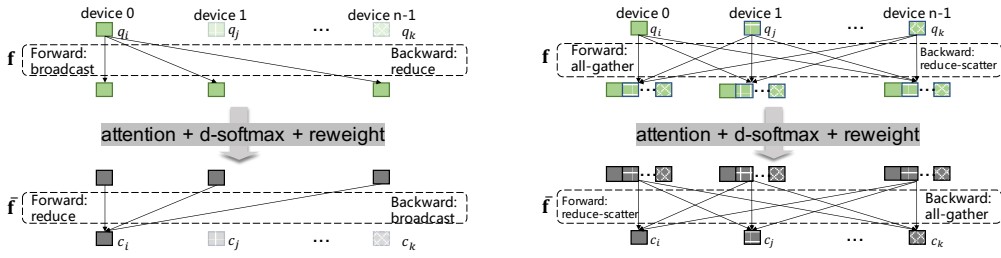

Figure 4: Conjugate pair comparisons. Broadcast/reduce: asymmetric simplex one-to-all communication for $n$ times. All-gather/reduce-scatter: symmetric duplex collective communication for once.

and concatenated $Q^j \in \mathbb{R}^{\frac{L}{m} \times d_k}$. Hence the distributed attention in a minor range could be conducted as:

$$C_i^j = \text{d-softmax}(S_i^j)V_i = \text{d-softmax}(\frac{Q^j K_i^\mathsf{T}}{\sqrt{d_k}})V_i \tag{5}$$

Where the $C_i^j$ is the sub-context matching $Q^j$ on the $i$-th device. The memory complexity of each micro-step's attention scores map $S_i^j$ would be $\mathbb{R}^{\frac{L}{m} \times \frac{L}{n}}$, illustrated as in Fig. 1c. The memory space could be reused across the micro-steps, implemented through the checkpointing technique (Chen et al., 2016). The existing methods typically employ layerwise checkpointing, not addressing the excess intermediate activations within a single layer, while MQSP fractionizes it to finer-grained partitions. At last, $\bar{\mathbf{f}}$ conducts reduce-scatter to sum up the sub-contexts $C_i^j$ back to their ranks, as the reduced micro-context $c_i^j$. Each rank concatenates micro-contexts to produce its local context $C_i$.

**Comparison.** Compared with $\mathbb{R}^{\frac{L}{n} \times L}$ in ColAISP, MQSP requires only $\mathbb{R}^{\frac{L}{m} \times \frac{L}{n}}$ memory space for the attention map. Coefficient $n$ along column axis and coefficient $m$ along row axis make joint efforts to disintegrate the quadratic memory of the Transformer, enhancing the scalability of sequence length.

In concern of communication, ColAISP transfers queries and keys in rings while MQSP collectively transfers queries and contexts, incurring comparable communication volumes. However, in the heterogeneous network environment, e.g., a cluster with 4 nodes × 8 GPUs with `Nvlink`, the ring-style ColAISP suffers from the bottleneck of the low inter-node bandwidth. In contrast, the duplex collective MQSP benefits from the hierarchical NCCL (Nvidia). Additionally, if ColAISP adopts the Micro-Q method to reduce memory overhead, the ring communication would be repeated $m$ times, unlike amortized in MQSP. Since the inputs and outputs are uncorrelated among the micro-steps, we could alleviate communication overhead by overlapping with the other step's computation, further boosting efficient distributed self-attention (Appx. A.3).

## 3.2 Distributed Softmax

This section introduces the distributed softmax with low cost in MQSP. A similar technique could be found in ArcFace (Deng et al., 2019) to recognize large-scale faces. ArcFace sums up the local denominator in the forward pass, while the backward formula is simplified with a cross-entropy loss. Here we analyze the general softmax. Defining the whole sequence of scores as $[s_0, s_1, ..., s_{L-1}]$ ($\sigma = \max(s_{i \in [0,L)})$) and the probabilities as $[p_0, p_1, ..., p_{L-1}]$, the original form is formulated as:

$$p_i = \frac{\exp(s_i - \sigma)}{\sum_{j=0}^{L-1} \exp(s_j - \sigma)}, \quad \nabla s_i == \nabla p_i \times p_i(1 - p_i) + \sum_{j \neq i}^{L-1} \nabla p_j \times (-p_i p_j) \tag{6}$$

In MQSP, the scores are distributed across devices, which incurs $O(\frac{n-1}{n}L)$ communications cost and $O(L)$ memory overhead in the form of Eq. 6. To this end, we convert the arithmetic form and hierarchically reduce the maximum or summation, to attain an efficient distributed softmax:

$$p_i = \frac{\theta_i}{\mathbf{r}_{sum}(\Theta_i)}, \quad \nabla s_i = \lambda_i - p_i \times \mathbf{r}_{sum}(\Lambda_i) \tag{7}$$

Where $\mathbf{r}_{sum}$ means the reduce-sum operation. $\theta_i = \exp(s_i - \sigma)$ and its local summation is $\Theta_i$. $\lambda_i = \nabla p_i \times p_i$ and its local summation is $\Lambda_i$. Please refer to Appx. A.2 for details of mathematical derivation. It introduces only $O(n-1)$ complexity to allreduce-sum the scalar of local summation, negligible compared with Eq. 6. The pseudo-code of the distributed softmax is given in Algo. 1.

---

**Algorithm 1** The forward and backward pass of the distributed softmax.

```
1 # s: local attention scores
2 #     [..., qDim, LocalSeqDim]
3 def d_softmax_forward(s):
4   max_local = s.max(-1)  # [..., qDim]
5   max_global = all_reduce(max_local, op=MAX)
6   s_exp = (s - max_global[..., None]).exp()
7   sum_local = s_exp.sum(-1)  # [..., qDim]
8   sum_global = all_reduce(sum_local, op=SUM)
9   p = s_exp / sum_global[..., None]
10  return p
```

```
1 # p: local attention probabilities
2 # p_grad: gradient of p
3 # both in [..., qDim, LocalSeqDim]
4 def d_softmax_backward(p, p_grad):
5   P = p_grad * p
6   sum_local = P.sum(-1)  # [..., qDim]
7   sum_global = all_reduce(sum_local, op=SUM)
8   s_grad = P - p * sum_global[..., None]
9   return s_grad
```

---

Table 1: Memory consumption comparison of model parameters and intermediate activations for the vanilla Transformer and the different parallel methods for the Transformer.

| METHOD | MODEL PARAMETERS | INTERMEDIATE ACTIVATIONS |
|---|---|---|
| VANILLA TRANSFORMER | $(8 + \frac{4}{H})D^2$ | $10BDL + 2BHL^2$ |
| MEGATRON-LM3 | $(8 + \frac{4}{H})\frac{D^2}{n}$ | $(2 + \frac{8}{n})BDL + 2BH\frac{L^2}{n}$ |
| COLAISP | $(8 + \frac{4}{H})D^2$ | $11BD\frac{L}{n} + 2BH\frac{L^2}{n}$ |
| MQSP | $(8 + \frac{4}{H})D^2$ | $10BD\frac{L}{n} + BD\frac{L}{m} + 2BH\frac{L^2}{mn}$ |

## 3.3 MEMORY USAGE ANALYSIS

This section analyzes the memory usage of the model parameters and intermediate activations in a single Transformer layer. We omit the gradients and optimizer states proportional to the model parameters and the statistical buffers or masks in dropout and layer normalization. $B, H, L, n$, and $m$ represent the batch size, multi-head size, sequence length, device number, and Micro-Q number, respectively. To be concise, we assume $D = d_m = Hd_v = Hd_k$, consistent with most implementations.

As for the model parameters, the linear layers in MLP take $8D^2$, and the $\mathcal{L}_{qkv}$ and output layer in self-attention take $\frac{4D^2}{H}$. It is the same for the vanilla transformer and sequence parallel methods, while Megatron-LM3 divides it by $n$. For the intermediate activation, the MLP takes $5BDL$, and the self-attention takes $5BLD + 2BHL^2$. MQSP divides the MLP part by $n$ and introduces the $\frac{1}{mn}$ factor in the attention map and an all-gathered Micro-Q buffer $BD\frac{L}{m}$. We apply the same memory analysis on Megatron-LM3 and ColAISP (Appx. A.4), as shown in Tbl. 1.

According to the analysis, tensor parallelism has memory superiority in model parameters, yet the intermediate activations consume most memory in the long sequence Transformer. The activations of previous methods include $L$ or $\frac{L^2}{n}$ factors, which limit their sequence scalability. Contrastively, MQSP adjusts the granularity of Micro-Q $m$, attaining efficient memory usage. Assuming the remained upper bound memory for intermediate activations as $\mathcal{M}$ and adjusting $m$ equivalent to $n$:

$$m = n, 10BD\frac{L}{n} + BD\frac{L}{m} + 2BH\frac{L^2}{mn} \leq \mathcal{M} \Rightarrow L \leq (\sqrt{(\frac{11D}{2H})^2 + \frac{\mathcal{M}}{2BH}} - \frac{11D}{2H})n = \mathcal{C}n \quad (8)$$

The maximum $L$ grows proportionally to $n$ with a constant $\mathcal{C}$, demonstrating the linear scalability of MQSP. It meets $\frac{L}{mn} \geq 1$ to ensure Micro-Q includes at least one query, which indicates Eq. 8 in the condition of $n \leq \mathcal{C}$, $L \leq \mathcal{C}^2$. Moreover, owing to the query granularity flexibility, we can set a larger $m$ to obtain finer-grained Micro-Q to save memory, attaining further sequence length scaling.

## 4 EVALUATIONS

This section evaluates the proposed MQSP, verifying its convergence quality and comparing it with other parallel methods in sequence length scalability, memory footprint, and throughput. We further investigate the influence of the Micro-Q setting. Specifically, we implement MQSP with Pytorch-1.9 (Paszke et al., 2019), referencing Bert (Kenton & Toutanova, 2019) implemented in the HuggingFace transformers (Wolf et al., 2020). The experimental hardware is a private cluster, each node containing Intel(R) Xeon(R) Platinum 8369B CPU, 760-GB of RAM, and eight 80-GB A100 GPUs with `Nvlink`, resulting in 96 times intra-node bandwidth compared with inter-node (300 GBps v.s. 25 Gbps).

## 4.1 QUALITY OF CONVERGENCE

MQSP distributes Transformer in sequence dimension to scale up long sequences with the full attention in the vanilla Transformer. To verify the convergence of MQSP, we experiment with our MQSP and distributed data parallel (DDP) on the datasets, including WikiText-103 (Stephen et al., 2017), SQuAD (Rajpurkar et al., 2016), QQP, and MRPC(Wang et al., 2018).

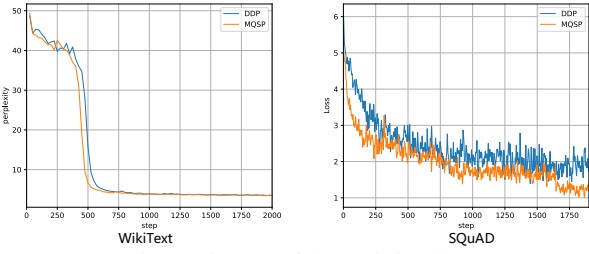

Figure 5: Part of the training loss.

| Tasks(Metric) | DDP | MQSP |
|---|---|---|
| WikiText(PPL) | 2.979 | 2.987 |
| SQuAD(F1) | 86.37 | 87.17 |
| QQP(F1) | 72.10 | 73.49 |
| MRPC(F1) | 86.38 | 87.99 |

Table 2: The convergence results.

We set the same parallel size as data parallel, e.g., $n = 8$ in one node, and identical training hyperparameters. The convergence results are shown in Tbl. 2, and the training loss curves are depicted in Fig. 5. The similar convergence quality demonstrates that MQSP provides a means for long sequences under the guarantee of convergence, which brings exploration space for Transformer toward longer sequences.

## 4.2 SEQUENCE LENGTH SCALABILITY

This subsection demonstrates the superior sequence length scalability of the proposed MQSP. We compare the maximum sequence lengths achieved by the Transformer using different distributed approaches as the number of devices increases.

We evaluate the methods by training the BERT-large model (Kenton & Toutanova, 2019). We use the batch size of 16 and the Adam optimizer following Kenton & Toutanova. Notably, because the checkpointing technique in the Micro-Q mechanism erases layers' stacked self-attention activations, we apply layerwise checkpointing to eliminate the accumulated activations across the Transformer layers for a fair comparison. Furthermore, for each particular number of devices $n$, we configure MQSP in two ways: a) MQSP-eq, where $m = n$ for linear scalability as analyzed in section 3.3, and b) MQSP-mem, where $m = L/n$ as the finest-grained Micro-Q to investigate the largest scaling potential. MQSP-mem serves as an upper bound reference, and for more about the Micro-Q setting, please refer to section 4.5.

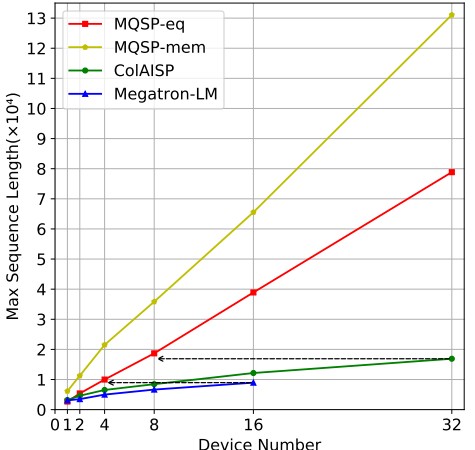

Figure 6: Comparison of maximum sequence lengths for training the Transformer as the GPU scale increases.

As shown in Fig. 6, the maximum sequence lengths of these methods are measured on $2^n$ GPUs, ranging from 1 to 32. The metric data range for Megatron-LM3 is limited to $2^4$ GPUs because the tensor parallel size must be able to divide multi-head dimension, which is 16 in BERT-large. In comparison, the scalability of the sequence parallel methods is less constrained.

We observe that MQSP-eq acquires $3.2\times$ and $4.3\times$ longer sequence than ColAISP and Megatron-LM3 (38912 v.s. 12160/8960) when $2^4$ GPUs, and $4.5\times$ than ColAISP (78848 v.s. 17408) when $2^5$ GPUs. MQSP-eq requires only a quarter of GPUs to achieve the maximum sequence of ColAISP and Megatron-LM3, denoted by the dotted arrow lines.

As the GPUs increase, Megatron-LM3 and ColAISP climb at a consistently decreasing rate, while MQSP-eq scales up almost linearly with a slope of $\mathcal{C} = 2464$. As analyzed in section 3.3, MQSP maintains linear scalability under the condition of $n \leq 2464$, $L \leq 2464^2$, which is practically a hardly attainable upper bound. It demonstrates that our MQSP resolves the quadratic memory overhead in the long sequence Transformer and achieves superior sequence length scalability. Moreover, MQSP-mem achieves further scalability, about $2\times$ compared with MQSP-eq, even on a single device (6144 v.s. 2688). It proves that the flexibility of the Micro-Q brings the potential to a more extended sequence, orthogonal to the sequence parallelism.

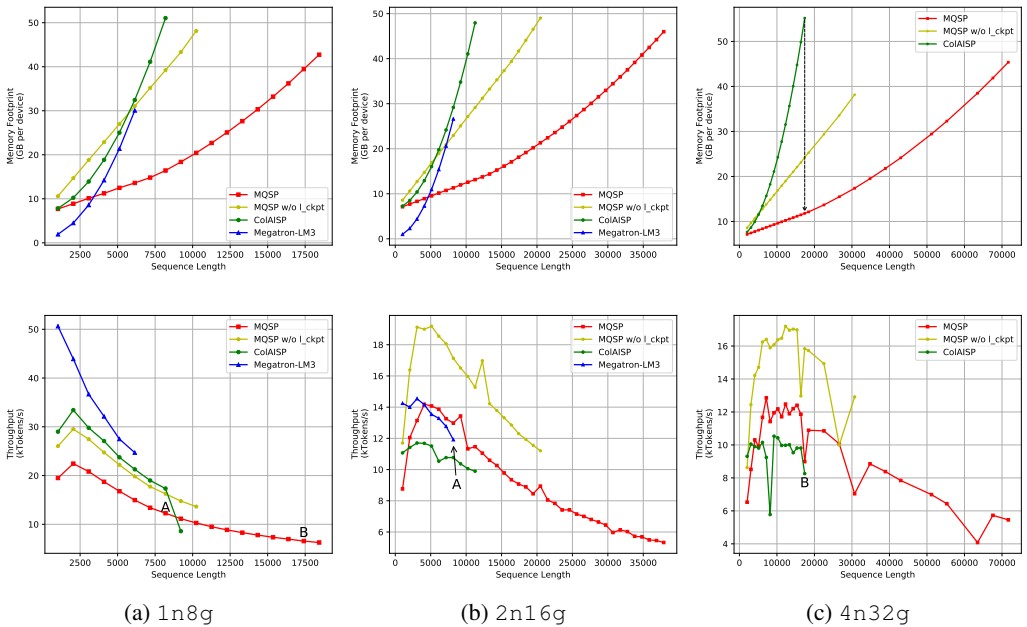

(a) `1n8g`         (b) `2n16g`         (c) `4n32g`

Figure 7: Comparison of the memory footprint and the token throughput with sequence length scaling up. `w/o l_ckpt` means without layer checkpointing.

### 4.3 MEMORY FOOTPRINT

In the sequence length experiment, MQSP achieves longer sequences with superior scalability, reflecting the advantage of MQSP's low memory footprint. Here, we conduct a more specific memory footprint comparison. Inheriting the model and training settings from the previous section, we arrange the experimental environments as `1n8g`, `2n16g`, and `4n32g`, where $XnYg$ represents $Y$-GPUs of $X$ nodes in our cluster. MQSP below represents MQSP-eq for linear scalability setting. In addition, since the checkpointing in Micro-Q erases most stacked activations, we also evaluate MQSP without layer checkpointing, `w/o l_ckpt` for short, to compare the memory and throughput. With the consistent hardware resources, we scales up the sequence length and compare their maximum allocated memory during training, as shown in the top row of Fig. 7.

Except for Megatron-LM3's advantage in short sequences with fewer model parameters, MQSP occupies less memory footprint than other methods on most configurations, saving up to 78.6% memory when 17408 on `4n32g`, as denoted by the dotted arrow line. Even with stacked activations, MQSP `w/o l_ckpt` also requires less memory in long sequences. It indicates that MQSP expects less memory to support the training of long sequence Transformer. Furthermore, the memory advantage of MQSP grows with longer sequences, benefiting from its advantageous memory efficiency in self-attention.

### 4.4 THROUGHPUT

**Token Throughput.** In addition to the capability for training longer sequence Transformer, training efficiency also deserves attention. We also conduct the token throughput comparison, as shown in the bottom row of Fig. 7. The throughputs generally decline with sequence length scaling up due to the quadratic computation complexity in self-attention. For the maximum sequence in different environments, MQSP achieves similar token throughput, demonstrating that MQSP could scale up $N\times$ sequence length with $N\times$ devices and $N\times$ time consumption.

In `1n8g`, Megatron-LM3 has throughput advantages, and MQSP `w/o l_ckpt` is comparable with ColAISP, while MQSP scales up to 18432 sequence without a significant drop in throughput. When training longer sequences, which are 8192 for Megatron-LM3 and 17408 for ColAISP, their insufficient scalability incurs the inter-node parallel group, marked as A and B in Fig. 7. Our MQSP supports the same length within one node, achieving 2.1× and 3.3× throughput per device to Megatron-LM3 and ColAISP, respectively. MQSP also has throughput advantages in multi-node environments, benefiting from efficient collective communications. In addition, MQSP `w/o l_ckpt`

gains further throughput with less recomputation while maintaining better sequence scalability than other methods.

**Time Ratios.** To analyze the throughput advantage of MQSP, we measure the time consumption ratios in a Transformer layer for MQSP and ColAISP. Configuring two environments as `1n4g` and `2n16g`, we train these two sequence parallel methods on their maximum sequence and profile the execution timeline of the forward passes. Here we adopt no overlapping to directly show the intra-node and inter-node communication costs. Fig. 8 exhibits the ratios of each part.

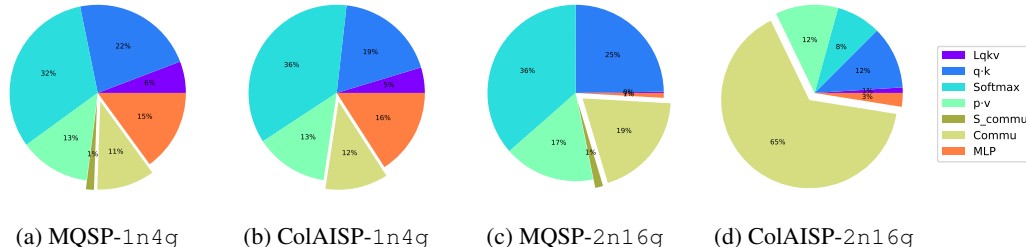

|  (a) MQSP-`1n4g` | (b) ColAISP-`1n4g` | (c) MQSP-`2n16g` | (d) ColAISP-`2n16g` |

Figure 8: Pie charts of time consumption with the communication part emphasized. The numeric suffix represents the number of nodes and GPUs. It indicates MQSP's efficient inter-node communication.

The order parts are the projection of $\mathcal{L}_{qkv}$, dot-production of $q$ and $k$, softmax or d-softmax, $v$ reweighting, d-softmax communications, ring or conjugate communications, and the MLP. It demonstrates that d-softmax introduces negligible communication costs. The communication costs occupy acceptably low ratios in intra-node scope for both MQSP and ColAISP, 11% and 12%, respectively. In an inter-node environment, the inadequate bandwidth between nodes results in different increased communication ratios, 19% for MQSP but 65% for ColAISP, as a sharper performance drop. It proves that the duplex collective communication adopted by MQSP brings an advantage in a heterogeneous network environment, compared with ColAISP's ring-style one restricted by the lowest link.

### 4.5 MICRO-Q SETTING

Micro-Q serves as the core part of the proposed MQSP, introducing the number $m$ of Micro-Q as a new hyperparameter. Thus this subsection explores the effect of how we set $m$. Training the BERT-large on 8192 sequence on `2n16g`, we vary $m$ to measure the maximum allocated memory and the throughput. As shown in Fig. 9, the memory gain saturates as $m$ reaches a certain level, for Micro-Q's attention computation is no longer where the maximum memory allocation occurs. Moreover, the throughput drops slowly as $m$ increases and then drops off rapidly after the memory gain saturates. The reason could be that excessive segmentation

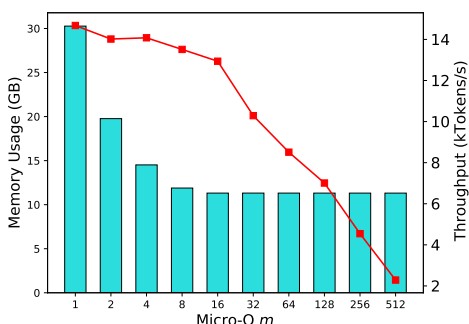

Figure 9: Memory (bar) and throughput (line) for different Micro-Q settings.

results in inefficient tiny computations and communications. The result indicates that we could make the trade-off between speed and memory, benefiting from the flexibility of the Micro-Q mechanism.

## 5 CONCLUSION

This paper presents the Micro-Query sequence parallelism, an efficient distributed method for linearly scaling long sequence Transformer. MQSP achieves distributed self-attention through all-gathering queries, maintaining only partial columns of attention map with a low-cost distributed softmax. Furtherly, MQSP introduces the finer-grained query, Micro-Q, to reuse memory among the rows of attention map, jointly decomposing the quadratic memory. MQSP attains 4.5× sequence length compared with ColAISP and 4.3× with Megatron-LM3, achieving up to 78848 sequence on 32 A100 GPUs. The flexibility of Micro-Q boosts further scalability orthogonally, even on a single device. MQSP saves 78.6% memory and achieves 3.3× speedup in memory and throughput evaluations. With guaranteed convergence, MQSP facilitates scaling longer sequence Transformer.

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

## A APPENDIX

### A.1 PSEUDO-CODE

**Communication Operators.**

**Algorithm 2** The pseudo-code of the conjugate communication operators.

```
1 class AllGatherQMicro(autograd.Function):
2   def forward(ctx, q_micro):
3     q = all_gather(q_micro)
4     return q
5   def backward(ctx, q_grad):
6     q_micro_grad = reduce_scatter(q_grad)
7     return q_micro_grad
```

```
1 class ReduceScatterC(autograd.Function):
2   def forward(ctx, c):
3     c_micro = reduce_scatter(c)
4     return c_micro
5   def backward(ctx, c_micro_grad):
6     c_grad = all_gather(c_micro_grad)
7     return c_grad
```

**Micro-Q Implementation.**

**Algorithm 3** The pseudo-code of the implementation of Micro-Q in self-attention.

```
1 class DistributedSelfAttention(nn.Module):
2   def forward(self, x):
3     Q, K, V = L_qkv(x)
4     c_micro_lst = []
5     for q_micro in Q.chunk(m):
6       c_micro = checkpoint(self.softmax_reweight, q_micro, K, V)
7       c_micro_lst.append(c_micro)
8     return cat(c_micro)
9   def softmax_reweight(self, q_micro, K, V):
10     Q_ = AllGatherQMicro.apply(q_micro)
11     attn_scores = matmul(_Q, K.T()) / sqrt(d_k)
12     attn_probs = DistributedSoftmax.apply(attn_scores)
13     C_ = matmul(attn_probs, V)
14     c_micro = ReduceScatterC.apply(C_)
15     return c_micro
```

### A.2 MATHEMATICAL DERIVATION OF DISTRIBUTED SOFTMAX

Consider an arbitrary row of the attention map of the scores and probabilities as $S_{row_k,:}, P_{row_k,:} \in \mathbb{R}^{1 \times L}$. To be specific, they include $[s_0, s_1, ..., s_{L-1}]$ and $[p_0, p_1, ..., p_{L-1}]$. For a numerically stable softmax, $s_i$ would minus their maximum value $\sigma = \max(s_{i \in [0,L)})$ before exponentiating:

$$p_i = \frac{\exp(s_i - \sigma)}{\sum_{j=0}^{L-1} \exp(s_j - \sigma)} \tag{9}$$

Mathematically, according to the derivative Jacobian matrix of softmax:

$$\frac{\partial p_i}{\partial s_i} = p_i(1 - p_i), \quad \frac{\partial p_i}{\partial s_j} = -p_i p_j (i \neq j) \tag{10}$$

For the training error $e$, the gradients for $s_i$ could be backpropagated as:

$$\nabla_e s_i = \sum_{j=0}^{L-1} \nabla_e p_j \times \frac{\partial p_j}{\partial s_i}$$
$$= \nabla_e p_i \times p_i(1 - p_i) + \sum_{j \neq i}^{L-1} \nabla_e p_j \times (-p_i p_j) \tag{11}$$

Specifically, we get each device's local maximum scores $\sigma_i = \max(s_{j \in [il, (i+1)l)})$ before communicating to collect $\sigma = \mathbf{r}_{max}(\sigma_i)$, where $\mathbf{r}_{max}$ means the allreduce-max operation. Defining the exponent of the normalized scores as $\theta_i = \exp(s_i - \sigma)$, we sum them up in stages:

$$\Theta_i = \sum_{j=il}^{(i+1)l-1} \theta_j, \quad p_i = \frac{\theta_i}{\mathbf{r}_{sum}(\Theta_i)} \tag{12}$$

Where $\mathbf{r}_{sum}$ means the reduce-sum operation. Similarly, for the backward pass, we define $\lambda_i = \nabla_e p_i \times p_i$ and its local summation $\Lambda_i = \sum_{j=il}^{(i+1)l-1} \lambda_j$. The form of backpropagation could be changed as:

$$\begin{aligned} \nabla_e s_i &= \nabla_e p_i \times p_i + \sum_{j=0}^{L-1} \nabla_e p_j \times (-p_i p_j) \\ &= \nabla_e p_i \times p_i - p_i \times \sum_{j=0}^{L-1} \nabla_e p_j \times p_j \\ &= \lambda_i - p_i \times \mathbf{r}_{sum}(\Lambda_i) \end{aligned} \tag{13}$$

### A.3 OVERLAPPING

As shown in Fig. 10, the computation of softmax on Micro-Q and values reweighting can overlap with other micro steps' communication of all-gather and reduce-scatter, offsetting the time overhead.

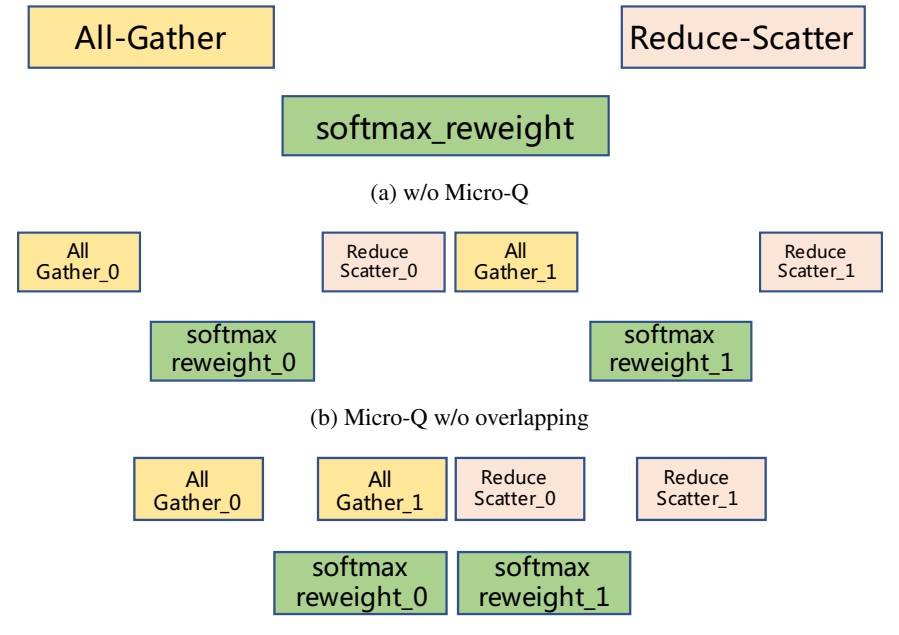

Figure 10: The overlapping of computation and communication.

### A.4 DETAILS ON MEMORY FOOTPRINT

$B, H, L, n,$ and $m$ represent the batch size, multi-head size, sequence length, device number, and Micro-Q number, respectively. Assume $D = d_m = Hd_v = Hd_k$.

**Vanilla Transformer.**

Model parameters:

- MLP: The first linear layer $d_m \times 4d_m$, and the second $4d_m \times d_m$.
- Self-attention: $\mathcal{L}_{qkv}$ layer $3 \times d_m \times Hd_k$, and output layer $Hd_v \times d_m$.
- Summation: $8d_m^2 + 3Hd_m d_k + Hd_m d_v = (8 + \frac{4}{H})D^2$

Intermediate activations:

- MLP: Input $BLd_m$, and intermediate $BL4d_m$.

- Self-attention: Input $BLd_m$, $\mathcal{L}_{qkv}$ produces $3 \times BHLd_k$, attention scores and probabilities $2 \times BHL^2$, and reweighted value $BHLd_v$.
- Summation:

$$5BLd_m + BLd_m + BHL(3d_k + d_v) + 2BHL^2 = 10BDL + 2BHL^2$$

**Megatron-LM3.**

Model parameters:

- MLP: The first linear layer $d_m \times 4\frac{d_m}{n}$, and the second $4\frac{d_m}{n} \times d_m$.
- Self-attention: $\mathcal{L}_{qkv}$ layer $3 \times d_m \times \frac{H}{n}d_k$, and output layer $\frac{H}{n}d_v \times d_m$.
- Summation: $\frac{8d_m^2}{n} + \frac{3Hd_md_k + Hd_md_v}{n} = (8 + \frac{4}{H})\frac{D^2}{n}$

Intermediate activations:

- MLP: Input $BLd_m$, and intermediate $BL4\frac{d_m}{n}$.
- Self-attention: Input $BLd_m$, $\mathcal{L}_{qkv}$ produces $3 \times B\frac{H}{n}Ld_k$, attention scores and probabilities $2 \times B\frac{H}{n}L^2$, and reweighted value $B\frac{H}{n}Ld_v$.
- Summation:

$$BLd_m + \frac{4BLd_m}{n} + BLd_m + \frac{BHL(3d_k + d_v) + 2BHL^2}{n} = (2 + \frac{8}{n})BDL + 2BH\frac{L^2}{n}$$

**ColAISP.**

Model parameters: Same as the Vanilla Transformer.

Intermediate activations:

- MLP: Input $B\frac{L}{n}d_m$, and intermediate $B\frac{L}{n}4d_m$.
- Self-attention: Input $B\frac{L}{n}d_m$, $\mathcal{L}_{qkv}$ produces $3 \times BH\frac{L}{n}d_k$, attention scores and probabilities $2 \times BH\frac{L}{n}L$, the ring buffer $BH\frac{L}{n}d_k$, and reweighted value $BH\frac{L}{n}d_v$.
- Summation:

$$\frac{5BLd_m}{n} + \frac{BLd_m + BHL(4d_k + d_v) + 2BHL^2}{n} = 11BD\frac{L}{n} + 2BH\frac{L^2}{n}$$

**MQSP.**

Model parameters: Same as the Vanilla Transformer.

Intermediate activations:

- MLP: Input $B\frac{L}{n}d_m$, and intermediate $B\frac{L}{n}4d_m$.
- Self-attention: Input $B\frac{L}{n}d_m$, $\mathcal{L}_{qkv}$ produces $3 \times BH\frac{L}{n}d_k$, attention scores and probabilities $2 \times BH\frac{L^2}{mn}$, the all-gather Micro-Q buffer $BH\frac{L}{m}d_k$, and reweighted value $BH\frac{L}{n}d_v$.
- Summation:

$$\frac{5BLd_m}{n} + \frac{BLd_m + BHL(3d_k + d_v)}{n} + \frac{BHLd_k}{m} + \frac{2BHL^2}{mn} = 10BD\frac{L}{n} + BD\frac{L}{m} + 2BH\frac{L^2}{mn}$$

