# OpenReview forum: "MQSP: Micro-Query Sequence Parallelism for Linearly Scaling Long Sequence Transformer"
_ICLR.cc/2023/Conference — Submitted to ICLR 2023_

### Official Review · Reviewer_nVbE · 2022-10-19

**Confidence:** 4
**Clarity, Quality, Novelty And Reproducibility:** OK
**Correctness:** 3
**Technical Novelty And Significance:** 2
**Empirical Novelty And Significance:** 2
**Recommendation:** 5

**Strength And Weaknesses:**

Strengths

With Micro-Query, MQSP scales up sequence length linearly and shows a pretty good performance improvement. Greatly increases the length of sequences that can be handled in Sequence Parallelism.
MQSP has a better communication pattern and significantly improved performance compared to previous sequential parallelism techniques.
This paper provides a thorough theoretical analysis, while comparing other methods including the most advanced Tensor Parallelism and Sequence Parallelism systems from theory and experiment.


Weakness - major concerns

In the evaluation, only Bert-Large was used for testing. Testing with more models with different sizes can make the results more convincing.
In Section 3.1 “MICRO -QUERY SEQUENCE PARALLELISM - Comparison”, a comparison is made with ColAISP only. If a comparison can be made with Tensor Parallelism, it can give the reader more information to choose between Tensor Parallelism and Sequence Parallelism
In Section 4.3 “MEMORY FOOTPRINT”, the paper shows the performance of MQSP with the activation checkpoint turned on and off, but does not specify whether the activation checkpoint is turned on in other implementations
It can be observed that within a sequence length of 7500, Megatron-LM3 has a clear performance advantage over MQSP, and there needs to be a more prominent expression in the paper to demonstrate why we need long sequences for training, and how long we really need them. One concern is that extra-long sequences can lead to excessive computational cost without actually bringing much gain effect.


Weakness - minor concerns

In Section 4.1 “QUALITY OF CONVERGENCE”, the setting of sequence length for MQSP is missing.
For some readers who have not been exposed to much tensor parallelism, the background of this paper is slightly brief. Consider adding more detailed background on Tensor Parallelism.
Some letter symbols are used in advance in the figure, such as "m" in figure one. Maybe highlight the meaning of these letters in the article or caption, it will be easier to understand the article.

**Summary Of The Paper:**

The paper proposed MQSP, an optimized sequence parallelism technique for long sequence transformer. Finally, the evaluation result shows that MQSP can scale to longer sequences. In long sequence scenarios, MQSP can significantly reduce memory occupation and have higher throughput compared to Tensor Parallelism and Sequence Parallelism.

**Summary Of The Review:**

Tend to reject; Please check my comments above for more information.

---

> ### Author Response · Authors · 2022-11-17
> **Author Response**
>
> Thank you for your diligent review.
>
> Q1: Testing with more models with different sizes can make the results more convincing.
>
> A1: Good suggestion. We currently adopt Bert-Large as a representative Transformer and would extend to different model sizes and other Transformers like GPT, etc.
>
> Q2: In Section 3.1, a comparison made with Tensor Parallelism can give more information.
>
> A2: Thanks for your helpful advice. In Section 3.1 we compare with ColAISP to mainly reveal our improvement upon the sequence parallelism, while the tensor parallelism could serve as an orthogonal combination member. We would consider making a comprehensive comparison with different parallelisms.
>
> Q3: Section 4.3 whether the activation checkpoint is turned on in other implementations.
>
> A3: As mentioned in Section 4.2, we apply layerwise checkpointing to eliminate the accumulated activations across the Transformer layers for a fair comparison. We follow this configuration thus the activation checkpoint is turned on in other implementations in Section 4.3.
>
> Q4: Within a sequence length of 7500, Megatron-LM3 has a clear performance advantage over MQSP.
>
> A4: On the same environment $\texttt{1n8g}$, Megatron-LM3 indeed has a performance advantage on its bearable sequence length. Yet our method could scale up to a much longer sequence length, and achieve better throughput with larger batch size benefitted from the less memory footprint.
>
> Q5: Why we need long sequences for training, and how long we really need them, with excessive computational cost.
>
> A5: Long-range modeling is a difficult but promising trend. The current models usually limit resolution from reality default, like UHD video over minutes and hours, and long sequence ability closes the gap. For example, Pathfinder-X/256 in LongRangeArena involves 16k/65k sequences.
>
> mQ6: In Section 4.1, the setting of sequence length for MQSP is missing; Consider adding more detailed background on Tensor Parallelism; Some letter symbols are used in advance in the figure.
>
> A6: Thanks for your careful review and detailed suggestion. In Section 4.1, the settings of sequence length are consistent with the original Bert for convergence verification. And we would revise the flaws in the background and symbols following your comments.

---

### Official Review · Reviewer_h8ct · 2022-10-31

**Confidence:** 4
**Correctness:** 3
**Technical Novelty And Significance:** 2
**Empirical Novelty And Significance:** 3
**Recommendation:** 3

**Clarity, Quality, Novelty And Reproducibility:**

Though the experiments look promising, the novelty is limited and the discussion and empirical comparison with the highly related work FlashAttention are not provided (details are provided in Weakness).

The paper is hard to follow. There is little description of Figures 3 and 4. I found the current main text loosely coupled with the figures, giving me some difficulty in comprehension. I encourage the authors to put more efforts in explaining the proposed algorithm in details especially for pages 4 and 5. Also, I suggest that authors need to ensure that all the symbols occurred in the paper have definitions and shapes such as $c_i^j$ and $C_i$.

It is not clear to me where 2BDL comes from for MEGATRON-LM3 in Table 1. By using the sequence parallelism introduced in (Korthikanti, V., et al., 2022), the memory costs for all the intermediate activations including dropout and layernorm can be reduced by n times.

I didn't run the code, so I do not have any data for reproducibility.

Korthikanti, V., Casper, J., Lym, S., McAfee, L., Andersch, M., Shoeybi, M., & Catanzaro, B. (2022). Reducing Activation Recomputation in Large Transformer Models. arXiv preprint arXiv:2205.05198.



**Strength And Weaknesses:**

Strength:
1. The empirical results are promising. Supporting the training with 78848 sequence length is impressive.

Weakness:
1. This paper lacks sufficient discussion and comparison with the related works. Similar idea of splitting sequence into smaller chunks and incrementally accumulating results has been proposed in (Dao, Tri, et al., 2022). This work investigates the bottleneck of Transformer training from IO perspective. The resultant algorithm fuses the whole Transformer layer into a fused kernel and the memory cost of computing attention map is constant independent of the sequence length. This algorithm has been implemented in multiple libraries including OpenAI triton and Xformers https://github.com/facebookresearch/xformers/blob/91c7c0846455d78b934d6705026b4c7e35b89bb1/xformers/ops.py#L280. We can easily combine it with the Megatron implementation.
2. Distributed softmax is also not new. This has been used in Megatron for computing output logits: https://github.com/NVIDIA/Megatron-LM/blob/main/megatron/mpu/cross_entropy.py

Dao, Tri, et al. "FlashAttention: Fast and Memory-Efficient Exact Attention with IO-Awareness." arXiv preprint arXiv:2205.14135 (2022).

**Summary Of The Paper:**

This work studies long sequence problem from the system perspective. The self-attention part of a Transformer requires quadratic memory cost, limiting the applicability of full-attention for thousands of tokens. There are major two lines of directions for tacking with the high complexity due to long sequence: 1) full-attention approximation such as sparse attention, and 2) model parallelism approach. This work focuses on the 2nd line. The existing works such as Megatron and ColAISP reduce memory complexity by n times, where n is the tensor parallel size. This paper proposed to further reduce the memory cost by splitting the attention computation into m micro-steps, and in each micro-step, a micro-context output is computed. In this way, the memory buffer can be reused and the memory cost of attention map is reduced by nm times. The experimental results show that the proposed method, dubbed MQSP, supports training a BERT-large model with 78848 sequence length on 32 A100 GPUs.

**Summary Of The Review:**

The paper studies an important problem since long sequence has a wide variety of applications such as question-answering and summarization. The empirical results are also promising and interesting. However, I am concerned with the novelty. The paper is lack of discussion and empirical comparison with FlashAttention. FlashAttention adopts very similar technique and fuses the computation of micro-chunks into a CUDA kernel, and its memory cost depends on the GPU SRAM size and is independent of the sequence length.

---

> ### Author Response · Authors · 2022-11-17
> **Author Response**
>
> Thank you for your constructive review.
>
> Q1: The paper is lack of discussion and empirical comparison with FlashAttention.
>
> A1: FlashAttention is a most recent brilliant work that erases the attention map from HBM with a highly-fused kernel and similar idea of chunking and recomputation. Though the motivation for amortizing the attention map would need to be reconsidered, our method could serve as an orthogonal means to FlashAttention. They are two different directions of thought for the same target, GPU kernel fusing and cross-GPU distributing for Transformer attention. The view of distributing brings with the device's dimension for computation ability scaling, And within the attention blocks of each device of sequence parallelism, the idea of reducing HBM memory access could also be applied for orthogonal optimization. We would further investigate and compare with FlashAttention.
>
> Q2: Distributed softmax is also not new. This has been used in Megatron for computing output logits.
>
> A2: Megatron computes output logits with cross-entropy to simplify the backward formula of distributed softmax. Though maybe not a new idea, we conduct distributed derivation for a solo softmax instead, serving sequence parallelism to amortize the attention map along the softmax dimension.
>
> Q3: There is little description of Figures 3 and 4; the main text loosely coupled with the figures; explaining the proposed algorithm in details; definitions of the symbols occurred.
>
> A3: Thanks for your careful review and detailed suggestion. We would revise the flaws in figures, symbols, and descriptions following your comments.
>
> Q4: It is not clear to me where 2BDL comes from for MEGATRON-LM3 in Table 1.
>
> A4: Details on memory footprint analysis could be found in Appendix.4. The 2BDL comes from the input tensors of MLP and self-attention, which are all-gathered along sequence dimensions to enter the tensor parallelism region.

---

### Official Review · Reviewer_zrYy · 2022-10-31

**Confidence:** 4
**Correctness:** 3
**Technical Novelty And Significance:** 2
**Empirical Novelty And Significance:** 2
**Recommendation:** 3

**Clarity, Quality, Novelty And Reproducibility:**

Many figures in the paper is not illustrative. I eventually figure out the following questions by

1. Figure 1: Which dimension corresponds to queries, and which dimension corresponds to keys?
2. Figure 2: I suppose $Q_i, K_i, V_i$ are all $L/n\times d_k$ sized matrices, but one cannot tell the $d_k$ dimension from the figure.
3. Figure 3: The size of $Q_i, K_i, V_i$ in the figure does not reflect their actual shape (which should be exactly the same!). Also $Q_i$ and $Q^j$ are of the same shape, which should not be the case.
4. Figure 3 caption: “The red dotted arrow lines represent the aligned row for distributed softmax.” Do you mean the orange dotted arrow? What does the arrow direction mean?

The paper should be able to be reproduced. The proposed method from the paper is new.

**Strength And Weaknesses:**

The paper proposed a more fine-grained method to partition the query dimension to reduce the memory footprint of the self-attention layer. The proposed method is new. However, I have doubts on the actual effectiveness of the method. Specifically,

1. The $O(L^2)$ memory usage for the attention operator can be reduced to $O(L)$ by changing the way it was implemented in GPU without introducing any communication overhead across different GPUs. See the following two references:
    1. [https://arxiv.org/pdf/2205.14135.pdf](https://arxiv.org/pdf/2205.14135.pdf)
    2. [https://arxiv.org/pdf/1911.02150.pdf](https://arxiv.org/pdf/1911.02150.pdf)

    Please compare with these works and show the effectiveness of the proposed method since the biggest benefit of the proposed does not exist any more with the optimized single-GPU kernels, and the proposed method brings much more extra communication.

2. There is no analytical analysis on the extra communication brought by the newly proposed algorithm, which is the major overhead brought by the proposed method.
3. Why do you need to evaluate the convergence of the proposed method in 4.1? The method does not change the behavior of the original self-attention algorithm.

**Summary Of The Paper:**

The paper proposed a new way to further reduce the memory footage of the self-attention mechanism over long sequences by parallelization on the sequence dimension. The paper proposed to communicate the query tensors to handle the sentence-dimension computation used in self-attention.

**Summary Of The Review:**

I have doubts about the effectiveness of the proposed method. I will raise my score if my concerns are addressed during the rebuttal.

---

> ### Author Response · Authors · 2022-11-17
> **Author Response**
>
> Thank you for your insightful review.
>
> Q1: Please compare with [FlashAttention, Fast Transformer Decoding] and show the effectiveness of the proposed method.
>
> A1: FlashAttention is a most recent brilliant work that erases the attention map from HBM with a highly-fused kernel. Though the motivation for amortizing the attention map would need to be reconsidered, our method could serve as an orthogonal means to FlashAttention. They are two different directions of thought for the same target, GPU kernel fusing and cross-GPU distributing for Transformer attention. The view of distributing brings with the device's dimension for computation ability scaling, And within the attention blocks of each device of sequence parallelism, the idea of reducing HBM memory access could also be applied for orthogonal optimization.
>
> The second reference improved Transformer decoding with shared K/V cache, and we would further investigate and compare with them.
>
> Q2: There is no analytical analysis on the extra communication brought by the newly proposed algorithm, which is the major overhead brought by the proposed method.
>
> A2: Our method collectively transfers queries and contexts, incurring comparable communication volumes to ColAISP. We would consider the analytical analysis of communication in further revising the paper.
>
> Q3: Why do you need to evaluate the convergence of the proposed method in 4.1? The method does not change the behavior of the original self-attention algorithm.
>
> A3: The method indeed computes the same original self-attention, and we evaluated the convergence to verify the implementation for excluding the precision difference from GEMM and tf32 type. We would consider rearranging the experiments relevant to performance on tasks.
>
> Q4: Many figures in the paper is not illustrative.
>
> A4: Thanks for your careful review. In Figure 1, the row dimension corresponds to queries and the column to keys; in Figure 2, the $d_k$dimension is omitted for brevity; in Figure 3, K/V are of the same shape as Q but turned around following Figure 2, and $Q_i$ and $Q^j$ are of the same shape only if $m == n$; in Figure 4, the red/orange dotted arrows mean that the softmax is conducted on those rows, while the arrow direction has no specific meaning. We would revise the flaws in the figures following your comment.

---

### Official Review · Reviewer_L8Ft · 2022-11-01

**Confidence:** 3
**Correctness:** 3
**Technical Novelty And Significance:** 3
**Empirical Novelty And Significance:** Not applicable
**Recommendation:** 5

**Clarity, Quality, Novelty And Reproducibility:**

This paper is mostly written clearly and the writing quality is good. As for originality, this is not the first work to propose sequence parallelism demonstrates superior scalability over the prior art leveraging sequence parallelism (ColAISP, in particular).  The idea of distributed softmax seemingly has been explored elsewhere.



**Strength And Weaknesses:**

**Strengths**

* The sequence length of transformers is likely to keep increasing, and a scalable solution to the sequence length has values.
* The proposed solution effectively increases the sequence length over 70K on 32 A100 GPUs.

** Weaknesses **

* It is not clear whether this technique introduces another bottleneck (especially with increasing $m$) for inter-GPU communication and synchronization.
* The paper lacks comparison against other works leveraging different forms of parallelism (e.g., pipeline parallelism + tensor parallelism).
* The necessity of scaling sequence length needs to be better motivated.


**Summary Of The Paper:**

This paper introduces a novel dimension to partition self-attention layers for distributed training of long-sequence transformers. The proposed technique, called Micro-Query Sequence Parallelism (MQSP), maintains only partial columns of attention map and applies a low-cost distributed softmax. To reuse memory over multiple rows of the attention map, MQSP also utilizies a finer-grained query, Micro-Q, to scale memory overhead linearly instead of quadratically. Evaluation demonstrates MQSP can scale sequence length significantly longer than both ColAISP and Megatron-LM3.


**Summary Of The Review:**

This paper focuses on improving scalability of multi-head self-attention block with an increasing sequence length. The self-attention block is known to be a scalability bottleneck due to it's quadratic overhead for both memory capacity and computation.  That said, this paper tackles a timely problem.  Evaluation demonstrates the sequence parallelism can handle longer sequences and is more memory space-efficient than both ColAISP (leveraging sequence parallelism in a different way) and Megatron (leveraging tensor parallelism).

While the results show some promise, I still have the following questions/concerns about this proposal:

* *Performance Sensitivity* - In Figure 9, there is a tradeoff between reduction in memory usage and token (training) throughput. This makes sense as $m$ increases, attention maps are partitioned into finer-grained chunks to increase the overhead of communication limiting overall throughput. In this particular setup of *2n16g*, the throughput is maintained at a high level before the benefits of memory savings saturate. However, this might not necessarily true depending on the hardware configuration (e.g., # of nodes, # of GPUs, GPU generations, communication bandwidth, etc.).  Would the proposed technique provide robust performance for other system configurations with different compute/communication ratios?
* *Comparison against Existing Work* - This work compares mostly against ColAISP, which also leverages sequence parallelism.  I was not aware of this work, and hence do not know how strong a baseline it is. Comparison against Megatron-LM3 alleviates the concerns but I am still wondering how MQSP compares against other existing works, especially leveraging multiple forms of parallelism (e.g., pipeline parallelism together with tensor parallelism) in terms of overall token throughput.
* *Motivation* - This work could have been better motivated by suggesting some realistic use cases requiring longer sequences. The authors suggest high-resolution image cases as an example, but wouldn't scaling down the input image be a more practical solution, especially for classification and segmentation tasks? What is the practical range of sequence lengths today (and how does it scale) and what kind of applications could potentially benefit from a long sequence with >78K tokens?

---

> ### Author Response · Authors · 2022-11-17
> **Author Response**
>
> Thank you for your detailed review.
>
> Q1: Performance Sensitivity. Would the proposed technique provide robust performance for other system configurations with different compute/communication ratios?
>
> A1: We have another experiment on 32 GPU machines with 16k sequence length, and also have the same conclusion: throughput will not significantly decrease before until the memory bottleneck is reached when micro_q increases. This is due to two reasons: when micro_q increases,  1. $O(\frac{L^2}{mn} )$ memory usage decreases but $O({L})$ becomes the main part, so memory bottleneck reach earlier; 2. GMM \(general matrix multiple\) block sizes will be decreased, we found multipart small GMM will not be significantly slower when matrix size is large than 1k*1k; Of course, there is room for further optimization of our methods such as fused kernel and activation memory usage manage, to make the throughput decline trend more slowly.
>
> Q2: Comparison against Existing Work. How MQSP compares against other existing works, especially leveraging multiple forms of parallelism?
>
> A2: It is a good suggestion to compare different parallelisms and their combinations. We currently compare ColAISP and Megatron-LM3, where sequence parallelism is involved for long sequence situations while pipeline parallelism are difficult to handle quadratic complexity. We would consider extending the experiments for a comprehensive comparison.
>
> Q3: Motivation. What kind of applications could potentially benefit from a long sequence with >78K tokens?
>
> A3: Long-range modeling is a difficult but promising trend. The current models usually limit resolution from reality default, like UHD video over minutes and hours, and long sequence ability closes the gap. For example, Pathfinder-X/256 in LongRangeArena involves 16k/65k sequences.
>
> Q4-1: As for originality, this is not the first work to propose sequence parallelism.
>
> A4-1: It indeed is not the first to parallel Transformer along sequence dimension. Yet we made improvements for sequence parallelism, saving 78.6% memory usage and achieving up to 3.3X in training efficiency.
>
> Q4-2: The idea of distributed softmax seemingly has been explored elsewhere.
>
> A4-2: Similar distributed softmax techniques have been explored in large-scale classification, vocabulary embeds slicing, and so on, yet mostly with cross-entropy to simplify the backward formula. Though maybe not a new idea, we conduct distributed derivation for a solo softmax instead, serving sequence parallelism to amortize the attention map along the softmax dimension.

---

### Decision · Program_Chairs · 2023-01-20

**Decision:**

Reject

**Justification For Why Not Higher Score:**

The paper currently suffers from the following weakness.
Weakness:
1. Missing comparisons to existing work on memory efficient attention.
2. Unclear motivation for considering long sequence lengths of O(78k).

**Justification For Why Not Lower Score:**

N/A

**Metareview: Summary, Strengths And Weaknesses:**

All reviewers find the proposed approach interesting. However they raised significant concerns about limited experimental results and comparisons with existing memory efficient attention methods. Further the paper is lacking in convincing applications for considering long sequence lengths of O(78k). Is there a setting of relevance where using such a long sequence length leads to better performance? Overall I believe the paper is currently ready for publication and addressing reviewers concerns can make the paper stronger.

Strengths: Novel approach to reducing attention cost for long sequence lengths ~ 78k.

Weakness:
1. Missing comparisons to existing work on memory efficient attention.
2. Unclear motivation for considering long sequence lengths of O(78k).